# Interplay of asymmetry and fragmentation in the many-body tunneling dynamics of two-dimensional bosonic Josephson junctions

Anal Bhowmik,[1, 2, *] Rhombik Roy,[3, 4] and Sudip Kumar Haldar[5, †]

[1] *Homer L. Dodge Department of Physics and Astronomy,*
*The University of Oklahoma, Norman, Oklahoma 73019, USA*
[2] *Center for Quantum Research and Technology, The University of Oklahoma, Norman, Oklahoma 73019, USA*
[3] *Department of Physics, University of Haifa, Haifa 3498838, Israel*
[4] *Haifa Research Center for Theoretical Physics and Astrophysics, University of Haifa, Haifa 3498838, Israel*
[5] *Department of Physics and Material Science and Engineering,*
*Jaypee Institute of Information Technology, Noida - 201304, India*
(Dated: April 27, 2025)

It is well known that the many-body tunneling of a bosonic condensate leads to (longitudinal) fragmentation along the tunneling direction. In this work, we prepare the initial ground state as a (transversely) fragmented system by introducing a barrier oriented orthogonally to the tunneling direction and allow it to tunnel through a two-dimensional longitudinally and transversely-asymmetric bosonic Josephson junctions. For a fixed barrier height, we find that the initial transversal fragmentation is essentially independent of the asymmetry along the tunneling direction but reduces when the asymmetry is oriented orthogonally to the junction. We investigate the interplay between the interference of fragmentations and asymmetry in the junction by analyzing the rate of density collapse in the survival probability, the uncertainty product, and the nontrivial dynamics of the occupation of the first excited orbital. The interference of fragmentations is quantified by the ratio between the reduction of transverse fragmentation and the development of longitudinal fragmentation. We show that asymmetry along the junction (orthogonal to the junction) delays (accelerates), compared to the symmetric potential, in obtaining the maximal interference of fragmentations. Notably, self-trapping opposes the interference, whereas a resonant tunneling condition enhances it. Overall, we demonstrate that the influence of asymmetry on the competition between longitudinal and transversal fragmentations, which together govern the macroscopic tunneling dynamics of interacting bosons, arises purely from the many-body effects and has no counterpart in the mean-field theory.

## I. INTRODUCTION

A Bose-Einstein condensate (BEC) in a double-well potential, commonly referred to as the bosonic Josephson junctions (BJJ) [1, 2], creates a paradigmatic system for exploring the physics of many-body quantum tunneling which is the underlying mechanism of several fundamental physical effects. Due to the non-linear interparticle interactions between bosons, the system displays various exotic dynamical behaviors, such as Josephson plasma oscillations [3, 4], macroscopic self-trapping [5, 6], and collapse and revival sequences [7]. Josephson effects have been extensively explored in various intricate systems, namely, fermionic superfluids [8, 9], spin-orbit coupled BECs [10, 11], polariton condensates [12, 13], spinor condensates [14], and supersolids [15]. Owing to the conceptual importance of the Josephson effects in a symmetric double-well potential and optical lattices, where the atomic motion is primarily confined in the lowest band [16–19], recent years have seen a growing interest in exploring the tunneling dynamics in asymmetric double-wells [20–26] and tilted optical lattices [27–32]. The asymmetric double-well potential is particularly intrigu-

ing as it can be viewed as the structural unit of a tilted optical lattice, providing a rich platform for investigating novel many-body quantum phenomena. The asymmetric double-well potential also provides a rich platform for studying advanced tunneling phenomena, including interband quantum tunneling [27, 30, 33] and resonantly enhanced tunneling [34, 35]. Resonantly enhanced tunneling arises when the ground state in the upper well becomes degenerate with the excited state in the lower well due to the asymmetry in the double well, resulting in a significant increase in tunneling rate. This phenomenon has been demonstrated in various contexts [28, 35–38], highlighting its broad applicability in advancing the understanding of tunneling pheomena.

The many-body Josephson dynamics of ultracold atoms inherently involves a gradual loss of coherence within the system. Consequently, capturing the true essence of Josephson dynamics in interacting systems necessitates going beyond the mean-field approximation. In this context, the concept of fragmentation becomes crucial. Fragmentation, by definition, occurs when the reduced one-body density matrix of the system develops more than one macroscopic eigenvalue, signaling a departure from a purely condensed state. The phenomenon of fragmentation plays a significant role in the many-body dynamics of ultracold systems and has been extensively studied theoretically [39–48] and experimentally [49–51]. Notably, initially condensed states are observed to de-

* anal.bhowmik-1@ou.edu
† sudip.haldar@mail.jiit.ac.in

velop fragmentation during Josephson dynamics, both in symmetric [19, 48, 52–56] and asymmetric [25, 57, 58] double-well potentials. These studies have demonstrated that fragmentation is not merely a secondary effect but an integral feature of the dynamics. Therefore, understanding the role of fragmentation is essential for an accurate and comprehensive description of Josephson dynamics in ultracold atomic systems.

In a recent study, we investigated the Josephson dynamics of initially fragmented states in a two-dimensional (2D) symmetric double-double-well potential, where double-wells were created along both the longitudinal and transverse directions [59]. Our findings reveal a strikingly rich dynamical behavior that goes beyond the realm and predictions of the mean-field theory. Unlike a condensed state, an initially fragmented state exhibits intricate tunneling dynamics arising from the interplay of different fragmentations. The interplay of fragmentations introduces new layers of complexity beyond the conventional understanding of Josephson tunneling of condensed systems. Furthermore, the many-body dynamics in multi-well systems could offer valuable insights for the emerging field of atomtronics [60–63], where precise control over tunneling dynamics and coherence properties is crucial for the design and implementation of quantum circuits and atomic quantum devices [64, 65].

In this study, we investigate the tunneling dynamics of fragmented states in an asymmetric two-dimensional bosonic Josephson junction (2D BJJ), where asymmetry is introduced along both the tunneling direction (longitudinal) and the direction orthogonal to tunneling (transversal). The different initial fragmented states are prepared in the left region of space along the transverse direction of the tunneling. By systematically varying the asymmetry along the tunneling direction, we examine phenomena such as self-trapping and resonantly enhanced tunneling for fragmented states. A key focus is on understanding how the asymmetry in the potential influences the coupling between longitudinal fragmentation, which develops during the tunneling process, and transversal fragmentation, which arises from the geometry of the initial trap. To this end, we analyze the rate of density collapse in the survival probability and the occupations of single-particle states. Additionally, we explore how the coupling between fragmentations affects the uncertainty product along the transverse direction. Our findings reveal the intricate interplay between longitudinal and transversal fragmentations which we call the interference of fragmentations and the significant role of asymmetry of the potential in shaping their coupling during tunneling. This study provides the robustness of interference of fragmentations with the asymmetry in 2D BJJs, offering a broader understanding of fragmentation effects in tunneling phenomena.

## II. THEORETICAL FRAMEWORK

The many-body Hamiltonian of $N$ interacting bosons in two spatial dimensions can be written as

$$\hat{H}(\mathbf{r}_1.\mathbf{r}_2,...,\mathbf{r}_N) = \sum_{j=1}^{N} \left[ \hat{T}(\mathbf{r}_j) + \hat{V}(\mathbf{r}_j) \right] + \sum_{j<k} \hat{W}(\mathbf{r}_j - \mathbf{r}_k).$$
(2.1)

Here, $\hat{T}(\mathbf{r})$ is the kinetic energy and $\hat{V}(\mathbf{r})$ represents the external trap potential. $W(\mathbf{r}_j - \mathbf{r}_k)$ is the pairwise interaction between the $j$-th and $k$-th bosons. We assume the inter-boson interaction is modeled as repulsive Gaussian function [66] with $W(\mathbf{r}_1 - \mathbf{r}_2) = \lambda_0 \dfrac{e^{-(\mathbf{r}_1 - \mathbf{r}_2)^2/2\sigma^2}}{2\pi\sigma^2}$ and $\sigma = 0.25\sqrt{\pi}$. $\lambda_0$ is the interaction strength and the interaction parameter, $\Lambda$, is scaled with the number of bosons, $N$, as $\Lambda = \lambda_0(N-1)$. Note that the choices of shape and strength of interaction between bosons do not qualitatively affect the physical phenomena described here. We employ the natural units, i.e., $\hbar = m = 1$. Throughout this work $\mathbf{r} = (x, y)$ and all quantities to be discussed are dimensionless.

We employ a comprehensive correlated many-body method, namely, the multiconfigurational time-dependent Hartree approach for bosons (MCTDHB) [19, 26, 67–79] method, as implemented in MCTDH-X package [80–82], to obtain in-principle numerically exact ground state and its out-of-equilibrium dynamics. The MCTDHB method employs the time-dependent variational principle for the ansatz which is a linear-combination of all possible configurations generated by distributing the $N$ bosons over $M$ time-adaptive orbitals. The ansatz of the MCTDHB wavefunction is described as [68]

$$|\Psi(t)\rangle = \sum_{\{\mathbf{n}\}} C_{\mathbf{n}}(t)|\mathbf{n}; t\rangle,$$
(2.2)

where $C_{\mathbf{n}}(t)$ is the expansion coefficients and $|\mathbf{n}; t\rangle = |n_1, n_2, ..., n_M; t\rangle$ is called a permanent. The number of time-dependent permanents $|\mathbf{n}; t\rangle$ is $\binom{N+M-1}{N}$. The method reduces to the well-known time-dependent Gross-Pitaevskii equation [83] while $M = 1$.

The many-body results, demonstrated in this work, have been obtained by using $M = 8$ time-adaptive orbitals and $N = 10$ bosons in the MCTDHB method. The convergence is examined with $M = 10$ time-adaptive orbitals (see the Supplemental Materials [84]). For the numerical solution we have applied a grid of $128 \times 128$ points in a box of size $[-10, 10) \times [-10, 10)$ with periodic boundary conditions. Convergence of the results with the number of grid points is confirmed with a grid of $256 \times 256$ points. Additional details are provided in [84].

## III.  QUANTITIES OF INTEREST

### A.  Transversal fragmentation of the initial states

The transversal fragmentation of the initial state is measured by the occupation of the first excited orbital which is ungerade orbital ($u$-orbital), details are discussed in the next section. For a fragmented system studied below, the occupation of $u$-orbital is determined from $\frac{n_2}{N}$.

### B.  Uncertainty product

To proceed we require the time-dependent uncertainty product along the $y$-direction, $U(t) = \frac{1}{N}\Delta_{\hat{Y}}^2(t) \times \frac{1}{N}\Delta_{\hat{P}_Y}^2(t)$, where $\frac{1}{N}\Delta_{\hat{Y}}^2(t)$ represents the position variance per particle, and $\frac{1}{N}\Delta_{\hat{P}_Y}^2(t)$ denotes the momentum variance per particle along the $y$-direction. The variance per particle of an observable $\hat{A}$ is determined using the expectation values of $\hat{A}$ and $\hat{A}^2$. The expectation value of $\hat{A} = \sum_{j=1}^N \hat{a}(r_j)$ depends solely on the one-body operators, while the expectation of $\hat{A}^2$, which involves both one- and two-body operators, can be expressed as, $\hat{A}^2 = \sum_{j=1}^N \hat{a}^2(r_j) + \sum_{j<k} 2\hat{a}(r_j)\hat{a}(r_k)$. The variance is then given by [85]

$$
\begin{aligned}
\frac{1}{N}\Delta_{\hat{A}}^2(t) =& \frac{1}{N}[\langle\Psi(t)|\hat{A}^2|\Psi(t)\rangle - \langle\Psi(t)|\hat{A}|\Psi(t)\rangle^2] \\
=& \frac{1}{N}\left\{ \sum_j n_j(t)\int d\mathbf{r}\phi_j^*(\mathbf{r};t)\hat{a}^2(\mathbf{r})\phi_j(\mathbf{r};t) \right. \\
&- \left[\sum_j n_j(t)\int d\mathbf{r}\phi_j^*(\mathbf{r};t)\hat{a}(\mathbf{r})\phi_j(\mathbf{r};t)\right]^2 \\
&+ \sum_{jpkq} \rho_{jpkq}(t)\left[\int d\mathbf{r}\phi_j^*(\mathbf{r};t)\hat{a}(\mathbf{r})\phi_k(\mathbf{r};t)\right] \\
&\left. \times \left[\int d\mathbf{r}\phi_p^*(\mathbf{r};t)\hat{a}(\mathbf{r})\phi_q(\mathbf{r};t)\right] \right\},
\end{aligned}
\tag{3.1}
$$

where $\{\phi_j(\mathbf{r};t)\}$ are the natural orbitals, $\{n_j(t)\}$ the natural occupations, and $\rho_{jpkq}(t)$ are the elements of the reduced two-particle density matrix, $\rho(\mathbf{r}_1, \mathbf{r}_2, \mathbf{r}_1', \mathbf{r}_2'; t) = \sum_{jpkq} \rho_{jpkq}(t)\phi_j^*(\mathbf{r}_1';t)\phi_p^*(\mathbf{r}_2';t) \ \phi_k(\mathbf{r}_1;t)\phi_q(\mathbf{r}_2;t)$. For one-body operators which are local in position space, the variance described in Eq. 3.1 reduces to [86]

$$
\begin{aligned}
\frac{1}{N}\Delta_{\hat{A}}^2(t) =& \int d\mathbf{r}\frac{\rho(\mathbf{r};t)}{N}\hat{a}^2(\mathbf{r}) - N\left[\int \frac{\rho(\mathbf{r};t)}{N}\hat{a}(\mathbf{r})\right]^2 \\
&+ \int d\mathbf{r}_1 d\mathbf{r}_2 \frac{\rho^{(2)}(\mathbf{r}_1, \mathbf{r}_2, \mathbf{r}_1, \mathbf{r}_2; t)}{N}a(\mathbf{r}_1)a(\mathbf{r}_2).
\end{aligned}
\tag{3.2}
$$

Here $\rho(\mathbf{r};t)$ is the density at a given time $t$. We note that the uncertainty product can be written in the center-of-mass (c.m.) coordinate as [87]

$$
\begin{aligned}
U(t) =& \frac{1}{N}\Delta_{\hat{Y}}^2(t) \times \frac{1}{N}\Delta_{\hat{P}_Y}^2(t) = \Delta_{\hat{Y}/N}^2(t) \times \Delta_{\hat{P}_Y}^2(t) \\
=& \Delta_{\hat{Y}_{\text{c.m.}}}^2(t) \times \Delta_{\hat{P}_{Y_{\text{c.m.}}}}^2(t)
\end{aligned}
\tag{3.3}
$$

and the commutation relation $[\hat{Y}_{\text{c.m.}}, \hat{P}_{Y_{\text{c.m.}}}] = i$ for any $N$.

### C.  Survival probability of the left side of space

The survival probability measures the fraction of the system remaining in the initial well at any given time during the tunneling process. In this work, we prepare the initial states in the left region of space. Consequently, the survival probability for the left side of the space is defined as:

$$
P(t) = \int\limits_{x=-\infty}^{0} \int\limits_{y=-\infty}^{+\infty} dxdy\frac{\rho(x, y; t)}{N}.
\tag{3.4}
$$

## IV.  RESULTS AND DISCUSSIONS

We split this section into two parts, namely, longitudinal asymmetry and transversal asymmetry. In the first part, we prepare the ground state in a 2D longitudinally asymmetric potential and then study the quench dynamics in a longitudinally asymmetric 2D BJJ. Similarly, in the second part, we focus on the preparation of the ground state in a 2D transversely asymmetric potential, followed by a discussion of the corresponding quench dynamics in the transversely asymmetric 2D BJJ. The quantities investigated here require detailed information on the time-dependent many-boson wavefunction, explicitly the density, the reduced one-particle density matrix, and the reduced two-particle density matrix.

### A.  Longitudinal asymmetry

#### 1.  Preparation of the ground state in the longitudinally-asymmetric potential

We prepare the ground state of the bosons in the left part of space of the longitudinally-asymmetric potential

using the trap $V_L(x, y)$ where

$$V_L(x, y) = \frac{1}{2}(x+2)^2 - C_x x + f_L(y), \qquad (4.1)$$

and $C_x$ is the asymmetry parameter along the longitudinal ($x$)-direction. Here, $f_L(y) = \frac{1}{2}y^2 + Ve^{-y^2/8}$ with $V$ is the barrier height along the $y$-direction. According to Eqs. 2.1 and 2.2, we can prepare various ground states depending on the choices of $C_x$ and $V$. Here $C_x = 0$ produces a symmetric potential.

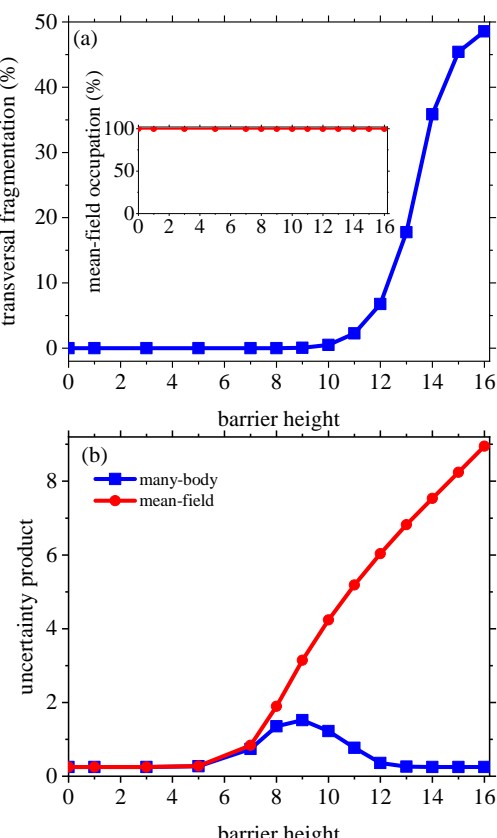

FIG. 1. Transversal fragmentation and uncertainty product along the $y$-direction in the symmetric 2D BJJ. (a) The many-body transversal fragmentations $n_2/N$ as a function of barrier height $V$. (b) Variation of the mean-field and many-body uncertainty products along the transverse direction $\frac{1}{N}\Delta_{\hat{Y}}^2 \frac{1}{N}\Delta_{\hat{P}_Y}^2$ with $V$. Inset of panel (a) shows the mean-field occupation of the ground orbital as a function of barrier height. The results of the initial condition are practically insensitive to the asymmetry $C_x$. We show here dimensionless quantities.

Initially, we prepare the ground state by gradually increasing the barrier height $V$ for both the symmetric potential and the asymmetric potential with asymmetry parameters $C_x$, $10C_x$, $20C_x$, and $25C_x$ with $C_x = 0.01$. As the barrier height $V$ divides the trap along the $y$-

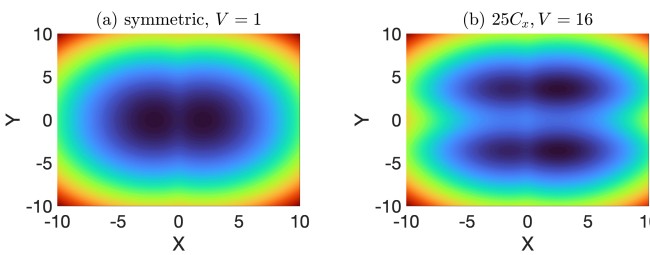

FIG. 2. External potential for tunneling dynamics in (a) the symmetric 2D BJJ with low barrier height $V = 1$ and (b) the longitudinally-asymmetric 2D BJJ (asymmetry $25C_x$ with $C_x = 0.01$) with high barrier height $V = 16$, described in Eq. 4.2. We show here dimensionless quantities.

direction, any fragmentation that develops in the system, due to geometry of the potential, is referred to as transversal fragmentation. Fig. 1(a) illustrates the transversal fragmentation for the symmetric potential as a function of $V$, quantified by the occupation of the first excited orbital, $\frac{n_2}{N} \times 100\%$. The geometry of the trap Eq. 4.1 dictates that the first orbital (ground orbital) is gerade ($g$-orbital) along the $y$-direction and the second orbital (first excited orbital) is ungerade ($u$-orbital) along the $y$-direction.

Let us begin by discussing the behavior of the system at the mean-field level of theory ($M = 1$). Within mean-field theory, only a single orbital, the ground orbital, is involved. As a result, the system remains fully condensed regardless of the barrier height or asymmetry in the potential, as indicated by the red solid circles in the inset of Fig. 1(a). In contrast to the mean-field theory, the many-body theory is capable of accurately presenting the ground state. According to the many-body results, the system remains fully condensed for $V = 0$ to 6 in the sense that less than 0.01% bosons are excited to the $u$-orbital. From $V = 7$ to 9, the system exhibits depletion, with less than 1% of bosons occupying the $u$-orbital, and eventually the system becomes fragmented for $V \geq 10$ with more than 1% of bosons occupy the $u$-orbital. Here, we use the term "fragmentation" broadly to refer to significant depletion rather than strictly to a macroscopic occupation of multiple natural orbitals. Notably, the many-body results reveal that the system becomes approximately 50% fragmented when the barrier height reaches $V = 16$, with the $g$-orbital and $u$-orbital almost equally populated. Additionally, the initial transversal fragmentation in the system is essentially independent of the longitudinal asymmetry parameter chosen here (therefore not shown). However, as we will see later, the dynamics exhibit intriguing physical behaviors depending on the longitudinal asymmetry in the potential.

Fig. 1(b) presents the mean-field and many-body uncertainty product along the $y$-direction, $U = \frac{1}{N^2}\Delta_{\hat{Y}}^2\Delta_{\hat{P}_Y}^2$. This quantity is particularly useful for the following reasons: (i) it is in principle highly sensitive to correlations, as it combines two variances, the position

variance $\frac{1}{N}\Delta_{\hat{Y}}^2$ and the momentum variance $\frac{1}{N}\Delta_{\hat{P}_Y}^2$ [87] and as we will see later (ii) it emphasizes the interference of fragmentations of many-body tunneling dynamics in BJJ. In the mean-field theory, $U$ monotonically increases with the barrier height, starting from an initial value of 0.25. This growth is attributed to the increasing spatial distribution of the ground state along the $y$-direction. In contrast to the mean-field result, the many-body $U$ increases up to $V = 9$ after which it gradually decreases and saturates back to about its initial value of 0.25. Fig. 1(b) clearly shows that the deviation between the mean-field and many-body results begins to grow at $V = 7$, corresponding to the onset of depletion in the ground state. Similar to the results for transversal fragmentation, $U$ is found to be practically independent of the asymmetry parameters in both the mean-field and many-body theories. Since the mean-field theory fails to qualitatively capture the system's behavior, at least for small numbers of bosons, we focus exclusively on the many-body results in the following discussions.

### 2. Tunneling of bosons in the longitudinally-asymmetric BJJ

To investigate the tunneling dynamics of initially prepared states, we solve the time-dependent Schrödinger equation, $\hat{H}(\mathbf{r}_1.\mathbf{r}_2,...,\mathbf{r}_N)\Psi(\mathbf{r}_1.\mathbf{r}_2,...,\mathbf{r}_N,t) = i\frac{\partial\Psi(\mathbf{r}_1,\mathbf{r}_2,...,\mathbf{r}_N,t)}{\partial t}$, for the scenario where the system evolves following a quench of the trap potential from $V_L(x,y)$ to the longitudinally-asymmetric 2D BJJ, $V_L'(x,y)$. The form of $V_L'(x,y)$, see Fig. 2, is

$$V_L'(x,y) = \begin{cases} \frac{1}{2}(x+2)^2 - C_x x + f_L(y) \\ \text{for } x < -\frac{1}{2}, -\infty < y < \infty, \\ \frac{3}{2}(1-x^2) - C_x x + f_L(y) \\ \text{for } |x| \leq \frac{1}{2}, -\infty < y < \infty, \\ \frac{1}{2}(x-2)^2 - C_x x + f_L(y) \\ \text{for } x > +\frac{1}{2}, -\infty < y < \infty. \end{cases} \quad (4.2)$$

Here, positive values of $C_x$ leads to the left part of space $(x < 0)$ being energetically higher.

*a. Many-body survival probability in the longitudinally-asymmetric BJJ:* In the tunneling dynamics, we begin our investigation with a basic quantity, the survival probability in the left side of space, $P(t)$. Fig. 3 illustrates the time evolution of $P(t)$ for the barrier heights $V = 1, 10, 11, 12, 13,$ and 16 (top to bottom). The first column shows the results for symmetric 2D BJJ. The asymmetry parameter is gradually increased from left to right with values $C_x$, $10C_x$, $20C_x$, and $25C_x$. For a fixed (longitudinal) asymmetry parameter, the frequency of oscillations of $P(t)$ is essentially the same for all barrier heights due to the practically identical Rabi frequency along the tunneling direction. In what follows, the time is scaled by the Rabi cycle, $t_{Rabi} = 132.498$, which is for symmetric 2D BJJ. Starting with the symmetric potential, we observe that the amplitude of $P(t)$ decays at different rates depending on the barrier height. At $V = 1$, the ground state is initially fully condensed, occupying only the $g$-orbital. During the tunneling dynamics, the amplitude of $P(t)$ gradually decays, eventually leading to a collapse of the density oscillations due to the emergence of longitudinal fragmentation. As the barrier height increases until $V = 6$, the decay rate of $P(t)$ decreases monotonically (not shown), even though no initial transversal fragmentation is present. This behavior occurs because the gradual increase in barrier height deforms the ground state along the transverse direction while preserving the system's coherence. Further increase of the barrier height $V \geq 7$ to $V = 12$, the initial ground state acquires transversal fragmentation which starts to couple with the longitudinal fragmentation developed during the tunneling process. This coupling between two fragmentations results in an increased decay rate of the amplitude of $P(t)$. Beyond $V = 12$, the initial transversal fragmentation and the tunnel-induced longitudinal fragmentation start to lose their coupling, leading to a delay in the collapse of the density oscillations, as found in [59].

Next, we analyze how the asymmetry in the BJJ impacts the tunneling of the ground states of different degree of fragmentations. For the asymmetry parameter $C_x$, we find that at the onset of the dynamics, approximately 60% of bosons tunnel back and forth between the left and right parts of space for all barrier heights. This behavior arises from the energy mismatch between the one-body ground states in the left and right regions of space. When the initial ground state is fully condensed, i.e., at $V = 1$, the amplitude of $P(t)$ starts to decay after about 12 Rabi cycle and continues to do so at a slower rate compared to the symmetric case. The slower rate of decay of the amplitude of $P(t)$ signifies a slower development of the longitudinal fragmentation. As the barrier height increases to $V = 10$ and 11, the decay in the amplitude of $P(t)$ is not observed within the considered time scale. However, at $V = 12$, we notice a slower decay rate of the amplitude of $P(t)$. Here, the initial transversal fragmentation accelerates the density collapse process for barrier heights ranging from $V = 7$ to $V = 12$. For $V > 12$ the coupling between transversal and longitudinal fragmentation weakens, thereby resisting the collapse of the density oscillations.

With an increase in the asymmetry parameter to $10C_x$, we observe that the energy mismatch between the one-body ground states in the left and right regions of space becomes significant enough to induce self-trapping of all bosons, regardless of the degree of fragmentation in the initial ground state. We find that the self-trapping region is in the vicinity of asymmetry parameter $10C_x$. Fur-

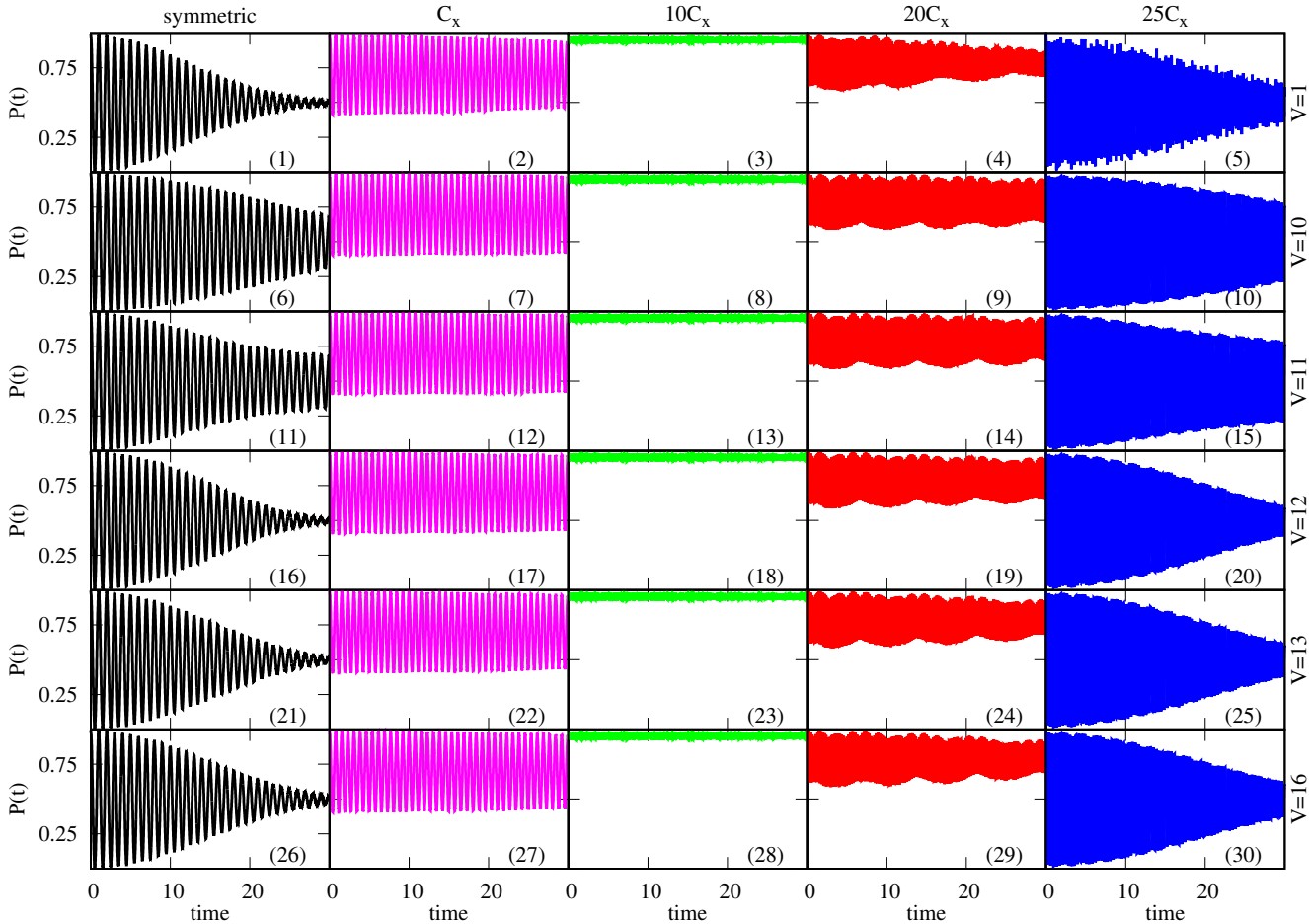

FIG. 3. Dynamics of the many-body survival probability of the left side of space, $P(t)$, for different barrier heights and asymmetry parameters in the longitudinally-asymmetric 2D BJJ. From the first to sixth rows present $P(t)$ for growing barrier heights, $V = 1, 10, 11, 12, 13,$ and $16$, respectively. The first column shows the results for symmetric BJJ. From second to fifth columns show $P(t)$ for growing asymmetry parameters, $C_x$, $10C_x$, $20C_x$, and $25C_x$, respectively, with $C_x = 0.01$. We show here dimensionless quantities.

ther increase of asymmetry, e.g., to $20C_x$, we notice that the initial states overcome the self-trapping region and the amplitude of $P(t)$ stabilizes at approximately 40%. However for $20C_x$, the collapse of density oscillation is the fastest for the low barrier height $V = 1$.

At the asymmetry parameter $25C_x$, we observe that, at the onset of tunneling dynamics, 100% of the bosons tunnel back and forth between the left and right regions of space. This asymmetry parameter is particularly intriguing because the ground state energy of the left region coincides with the first excited state energy of the right region for all barrier heights. This phenomenon, known as resonant tunneling, occurs between $\Psi$ of the left and $x\Psi$ of the right parts of space. Notably, when the initial state is fully condensed, i.e., at $V = 1$, and at the intermediate barrier height $V = 12$, the collapse of the density oscillations is fastest, resembling the behavior observed in the symmetric potential case. Moreover, it is found that for a fixed barrier height, the density collapse is slower, due to the slower development of longitudinal fragmen-

tation, for the resonant tunneling condition compared to the symmetric case

b. *Dynamic occupancy of orbitals and interference of fragmentations in the longitudinally-asymmetric BJJ:* To deepen the understanding of tunneling dynamics of fragmented BECs, it is crucial to analyze the dynamic occupancy of the first two orbitals. Note that for the longitudinally-asymmetric 2D BJJ, $y \to -y$ is a good symmetry. The first orbital, which remains the $g$-orbital, exhibits a monotonic decrease in occupancy during the tunneling dynamics, shown in [84]. This behavior occurs at varying rates depending on the barrier heights and asymmetry parameters. The reduction in the occupancy of the $g$-orbital is a natural consequence of the excitation of bosons to higher orbitals during the tunneling process. This analysis provides valuable insights into the redistribution of bosons among orbitals and highlights the dynamic nature of fragmented condensates under different potential configurations.

In contrast to the monotonic decrease observed in the

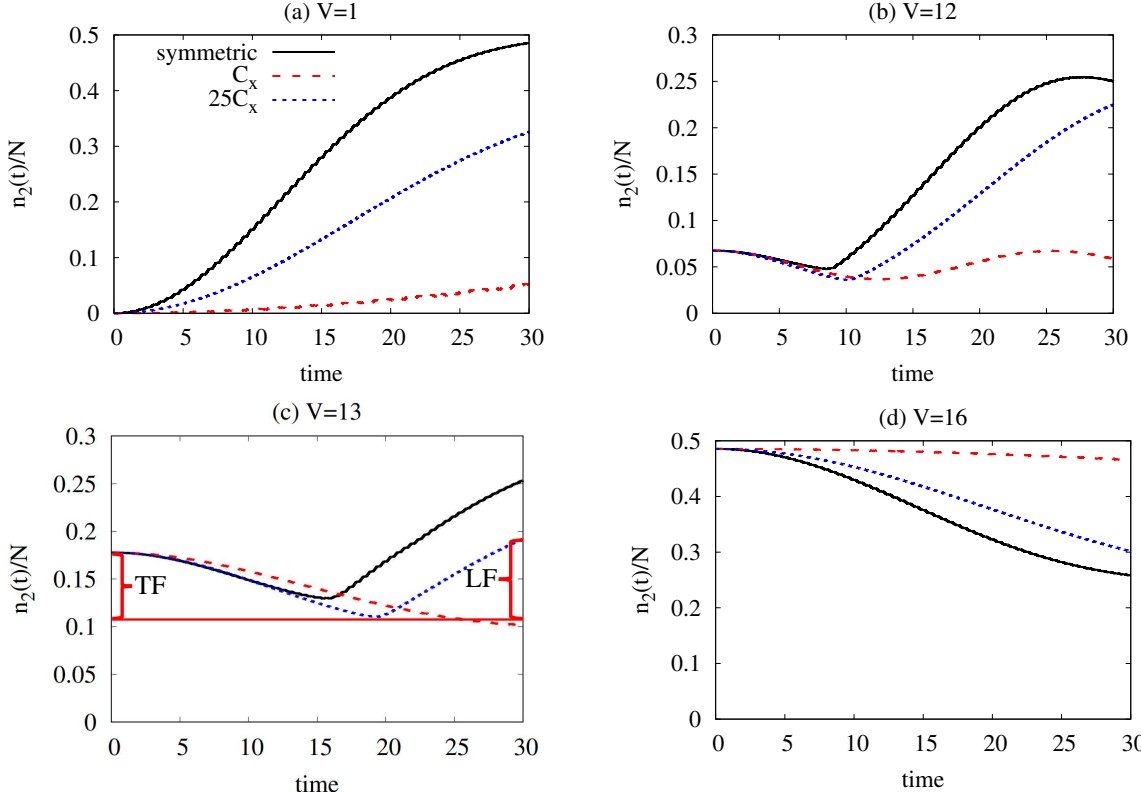

FIG. 4. Time evolution of the occupancy of the first excited orbital, $n_2(t)/N$, for the barrier heights (a) $V = 1$, (b) $V = 12$, (c) $V = 13$, and (d) $V = 16$, in the longitudinally-asymmetric 2D BJJ. The results are shown for symmetric BJJ and asymmetric BJJ with selective asymmetric parameters $C_x$ and $25C_x$, with $C_x = 0.01$. The monotonously increasing (decreasing) occupancy in the first excited orbital represents longitudinal fragmentation is strong (weak) compared to transversal fragmentation see for low barrier height $V = 1$ (high barrier height $V = 16$). In the intermediate barrier heights, see panels (b) and (c), $n_2(t)/N$ generally decreases until it reaches a point of minimum. Red solid line parallel to the $x$-axis represents a reference line which touches the point of minimum for asymmetry parameter $25C_x$. The magnitude of reduction of transversal fragmentation is denoted by TF and development of the longitudinal fragmentations is presented as LF, respectively. Color codes are explained in panel (a). We show here dimensionless quantities.

occupancy of the first orbital, the occupancy of the second orbital displays a more intricate behavior. Fig. 4 presents the occupancy of the second orbital for selective barrier heights ($V = 1$, 12, 13, and 16) and, symmetric and asymmetric potential (with asymmetry parameters $C_x$ and $25C_x$). These specific choices provide a representative overview of the general trends in the second orbital occupancy. As discussed in Fig. 1(a), at $t = 0$, the system is fully condensed for $V = 1$ to 6 (no excited orbital is initially occupied), and as the system is fragmented for $V \geq 7$, the first excited orbital is identified as the $u$-orbital. Note that, in the dynamics, for the fully condensed system, the second natural orbital is an excited $g$-orbital and for the fragmented system it is a $u$-orbital. Now, let us look at Fig. 4(a), i.e. for $V = 1$. Here the bosons gradually occupy the excited $g$-orbital due to the developed longitudinal fragmentation in the tunneling dynamics for all asymmetry parameters and the occupancy of the excited $g$-orbital monotonously grows. The rate of growth of occupancy of the excited

$g$-orbital is maximal for the symmetric potential, and decreases until the self trapping occurs. Further enlarging the asymmetry parameter, increases the rate of growth of occupancy of the excited $g$-orbital until the resonant condition is reached.

For partially fragmented systems, such as those with barrier heights $V = 12$ (Fig. 4(b)) and $V = 13$ (Fig. 4(c)), the occupancy of the first excited orbital, $u$-orbital, exhibits a complex temporal evolution. Initially, the occupancy decreases with time until it reaches a point of minimum (POM), after which it begins to increase. This transition from a reduction in occupation to an accumulation of occupation in the $u$-orbital is attributed to the interplay between the initial transverse fragmentation and the longitudinal fragmentation that develops during the tunneling dynamics. This behavior is observed for intermediate barrier heights $V = 7$ to 14 across all asymmetry parameters, except in the regime where self-trapping occurs. In the self-trapping condition, bosons are localized and do not exhibit tunneling dynamics, thereby prevent-

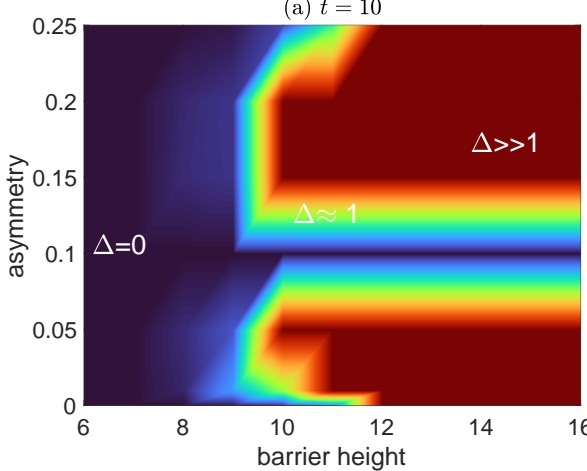

(a) $t = 10$

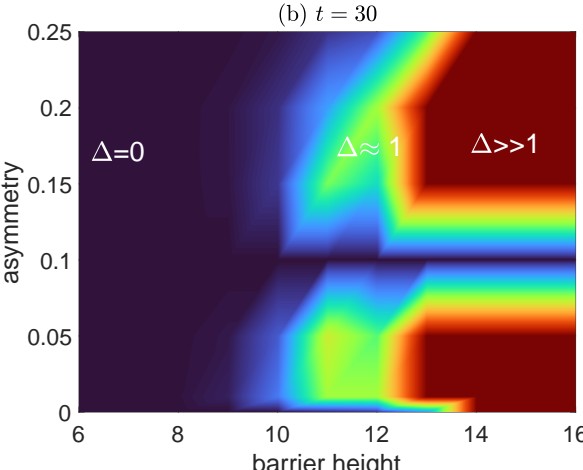

(b) $t = 30$

FIG. 5. Interference of transversal and longitudinal fragmentations, $\Delta$, is calculated from the ratio between the reduction of the transversal fragmentation and the development of longitudinal fragmentations $\Delta = \dfrac{\text{TF}}{\text{LF}}$ as a function of barrier height $V$ and asymmetry parameter $C_x$ at time (a) $t = 10$ and (b) $t = 30$. TF and LF are explicitly shown in Fig. 4(c). The results are shown for the longitudinally-asymmetric 2D BJJ. We show here dimensionless quantities.

ing the occupation of higher orbitals. A comparison of Figs. 4(b) and (c) indicates that the POM is delayed as the barrier height increases, for a fixed asymmetry parameter. Furthermore, for a given barrier height, the time required to reach the POM increases as the asymmetry parameter in the junction is raised. For instance, at $V = 13$, the POM is not reached for $C_x$ within the time window considered. Increasing the asymmetry in the junction further delays the POM until the system enters the self-trapping regime (e.g., for $10C_x$). However, beyond this point, as the asymmetry increases further, the time to reach the POM decreases, particularly when the resonant condition is satisfied (e.g., for $25C_x$). This detailed analysis highlights the critical role of the interference of transversal and longitudinal fragmentations in

determining the dynamics of orbital occupancy in fragmented BECs [59]. The observed delays and transitions at the POM are governed by the complex coupling between the longitudinal and transversal fragmentations, as modulated by barrier height and asymmetry in the system.

In Fig. 4(c), for $25C_x$, we explicitly demonstrate the reduction of transverse fragmentation (TF) and the build-up of longitudinal fragmentation (LF). TF is quantified by the difference between the initial occupation of the $u$-orbital at $t = 0$ and the occupation at the POM. Conversely, the longitudinal fragmentation (LF) is determined by the difference between the occupation of the $u$-orbital at time $t$ (after the POM) and the occupation at the POM. We define the ratio of TF to LF, $\Delta = \dfrac{\text{TF}}{\text{LF}}$, as a measure of the interference between the transversal and longitudinal fragmentations. From the analysis in Figs. 4(b) and (c), it is evident that $\Delta$ is time-dependent, reflecting the dynamic interplay between the two fragmentations. The detailed evolution of $\Delta$ and its implications will be discussed later.

For the fully fragmented system at $V = 16$, as shown in Fig. 4(d), the occupation of the $u$-orbital exhibits a monotonic decrease with an oscillatory background across all asymmetry parameters. This behavior closely mirrors the trend observed in the corresponding occupation of the $g$-orbital. It indicates that, in a fully fragmented system, the interference between longitudinal and transversal fragmentations does not develop during the tunneling process. Moreover, the ground state in this regime effectively behaves as two independent BECs, each containing $N/2$ bosons. These independent condensates, with no connection between them, are tunneling back and forth between the left and right spatial regions without any significant interaction or coupling between them.

To quantify the interference of the transversal and longitudinal fragmentations during the tunneling dynamics in the longitudinally asymmetric 2D BJJ, we plot $\Delta$ as a function of the barrier height $(V)$ and asymmetry parameter $(C_x)$ in Fig. 5 for times $t = 10$ and $t = 30$. Here, $\Delta = 0$ indicates one of two scenarios: either the system lacks initial transversal fragmentation and the tunneling dynamics involves only longitudinal fragmentation, or the system is self-trapped with initial transversal fragmentation (as observed near $10C_x$ in Fig. 5). A value of $\Delta$ corresponds to maximal interference of the transversal and longitudinal fragmentations, implying an approximately equal contribution from TF and LF. In contrast, $\Delta \gg 1$ signifies the dominance of transversal fragmentation over longitudinal fragmentation, where the occupation of the $u$-orbital does not reach the POM during the tunneling dynamics. Fig. 5 reveals that $\Delta$ evolves over time, particularly with an expansion of the region where $\Delta \approx 1$, which indicates stronger interference of fragmentations. Concurrently, the region where $\Delta \gg 1$ diminishes. This shift occurs due to the delayed POM in

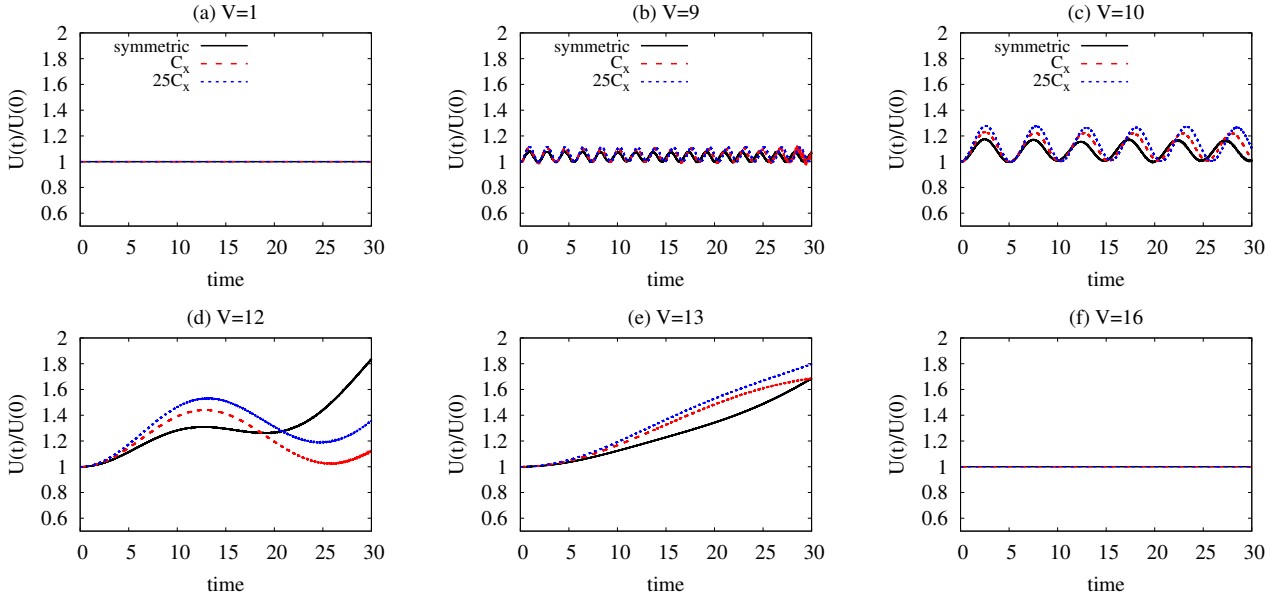

FIG. 6. Time evolution of the many-body normalized uncertainty product along the $y$-direction, $U(t)/U(0)$, in the longitudinally-asymmetric 2D BJJ. The barrier heights are (a) $V = 1$, (b) $V = 9$, (c) $V = 10$, (d) $V = 12$, (e) $V = 13$, and (f) $V = 16$. In each panel, $U(t)/U(0)$ is shown for the symmetric BJJ and asymmetric BJJs with asymmetry parameters, $C_x$ and $25C_x$ with $C_x = 0.01$. Color codes are explained in the top row. We show here dimensionless quantities.

the $u$-orbital occupation with increasing barrier height. These observations emphasize the role of asymmetry and barrier height on the interference of transversal and longitudinal fragmentations and its time-dependent nature.

*c. Dynamics of many-body uncertainty product in the longitudinally-asymmetric BJJ:* We now examine how the interference of longitudinal and transversal fragmentations influences a fundamental quantum mechanical quantity, the uncertainty product along the $y$-direction, $U(t) = \frac{1}{N^2}\Delta_{\hat{Y}}^2(t)\Delta_{\hat{P}_Y}^2(t)$. The analysis considers various asymmetries along the longitudinal direction and barrier heights. Fig. 6 illustrates the normalized uncertainty product, $U(t)/U(0)$, for a range of barrier heights, $V = 1, 9, 10, 12, 13,$ and 16. For each barrier height, we present results for both the symmetric potential and longitudinally asymmetric cases, corresponding to asymmetries $C_x$ and $25C_x$. This approach provides insights into the interplay between the barrier height and asymmetry in the longitudinal direction and their effects on the quantum uncertainty dynamics along the $y$-direction.

For $V = 1$, the initial state is fully condensed for both symmetric and asymmetric 2D BJJs, and only longitudinal fragmentation develops during the tunneling process. Since there is no competition between longitudinal and transversal fragmentations, $U(t)/U(0)$ remains essentially constant over time, as shown in Fig. 6(a). As the barrier height increases, the system transitions into a transversely fragmented initial state. The interplay between the initial transversal fragmentation and the longitudinal fragmentation developed during tunneling introduces oscillatory behavior in $U(t)/U(0)$, with a constant

amplitude as evident in Figs. 6(b) and (c). The amplitude of the oscillations increases with the asymmetry, highlighting the role of asymmetry in enhancing the coupling dynamics. For higher barrier heights ($V = 12$ and $V = 13$), $U(t)/U(0)$ exhibits a growing trend over time. This growth indicates a strong interference of longitudinal and transversal fragmentations. Beyond $V = 13$, $U(t)/U(0)$ gradually approaches a nearly frozen dynamics, signifying the dominance of transversal fragmentation over longitudinal fragmentation. At $V = 16$, the dynamics of $U(t)/U(0)$ stabilizes, showing an almost constant behavior for both symmetric and asymmetric BJJs. From these observations, we draw the following conclusions: (i) when either longitudinal or transversal fragmentation is dominant, $U(t)/U(0)$ exhibits a nearly frozen dynamical behavior, (ii) when transversal and longitudinal fragmentations compete, $U(t)/U(0)$ displays oscillatory dynamics, and (iii) the behavior of $U(t)/U(0)$ across the junction encodes information about the interference of longitudinal and transversal fragmentations.

By investigating the tunneling dynamics of different fragmented states in a longitudinally asymmetric 2D BJJ, we find that, while the initial states remain practically unchanged with varying asymmetry, their subsequent dynamics exhibit fundamentally different behaviors depending on the specific asymmetry introduced. In particular, the presence of initial transversal fragmentation significantly alters the tunneling dynamics whenever a substantial degree of longitudinal fragmentation develops during the evolution, e.g., for the symmetric potential and under resonant tunneling condition. A detailed analysis of the dynamic occupation of the orbitals

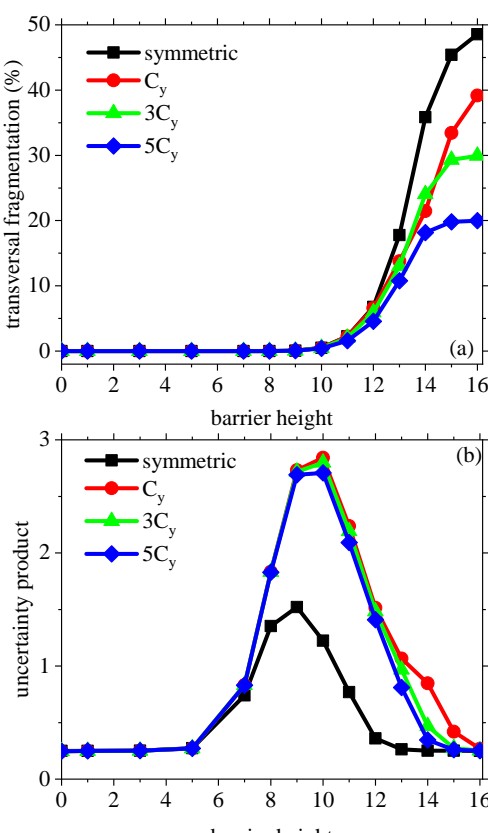

FIG. 7. Transversal fragmentation and uncertainty product along the $y$-direction in the transversely-asymmetric 2D BJJ. (a) The many-body transversal fragmentations $n_2/N$ as a function of barrier height $V$. (b) Variation of the many-body uncertainty product along the transverse direction $\frac{1}{N}\Delta^2_{\hat{Y}}\frac{1}{N}\Delta^2_{\hat{P}_Y}$ with $V$. The results are shown for the symmetric BJJ and the asymmetric BJJs with asymmetric parameter $C_y$, $3C_y$, and $5C_y$ with $C_y = 0.0001$. We show here dimensionless quantities.

reveals a competition between longitudinal and transverse fragmentations, influencing the overall tunneling process. Furthermore, examining the uncertainty product confirms that the interference of fragmentations has a measurable impact on key quantum mechanical observables, highlighting the intricate interplay of correlations in the many-body tunneling dynamics.

So far, we have explored how asymmetry along the tunneling direction influences the development of longitudinal fragmentation and its subsequent impact on the dynamics in a 2D BJJ. We now extend our investigation to examine whether asymmetry orthogonal to the tunneling direction induces similar tunneling dynamics or leads to a fundamentally different behavior.

### B. Transversal asymmetry

#### 1. Preparation of the ground state in the transversely-asymmetric potential

To prepare the ground state in a transversely-asymmetric potential, we introduce a linear slope along the $y$-direction. The transversely-asymmetric potential is then expressed as

$$V_T(x,y) = \frac{1}{2}(x+2)^2 + f_T(y), \qquad (4.3)$$

where $f_T(y) = \frac{1}{2}y^2 - C_y y + V e^{-y^2/8}$. Here $C_y$ is the asymmetry parameter along the transverse ($y$)-direction. $C_y = 0$ refers to the symmetric potential. Similar to the previous section, we begin our investigation with the initial properties of the ground state for a fixed asymmetry as a function of barrier height. In Fig. 7(a), we plot the transversal fragmentation, $\frac{n_2}{N} \times 100\%$ for the symmetric and transversely-asymmetric potential. The selected asymmetry parameters are $C_y$, $3C_y$, and $5C_y$, with $C_y = 0.0001$. Here we observe that the asymmetry in the potential, in general, opposes the development of the transversal fragmentation in the initial system. For instance, at the maximal barrier height $V = 16$, the system becomes almost 50% fragmented for symmetric potential, while it becomes about 40%, 30%, and 20% fragmented for asymmetry values $C_y$, $3C_y$, and $5C_y$, respectively.

Fig. 7(b) presents the uncertainty product along the $y$-direction, $U = \frac{1}{N^2}\Delta^2_{\hat{Y}}\Delta^2_{\hat{P}_Y}$, for the initial ground states as a function of the barrier height for symmetric potential and the three selected asymmetry values $C_y$, $3C_y$, and $5C_y$. As observed in the symmetric case and for all asymmetry parameters, $U$ initially increases with the barrier height, reaches a maximum, and then gradually decreases until it returns to about its initial value of 0.25. The plot indicates that higher asymmetry causes the system to return to the initial value of $U$ at smaller barrier heights.

#### 2. Tunneling of bosons in the transversely-asymmetric BJJ

In order to investigate the tunneling dynamics, we quench the potential from $V_T(x,y)$ to $V'_T(x,y)$. The transversely-asymmetric 2D BJJ is given by

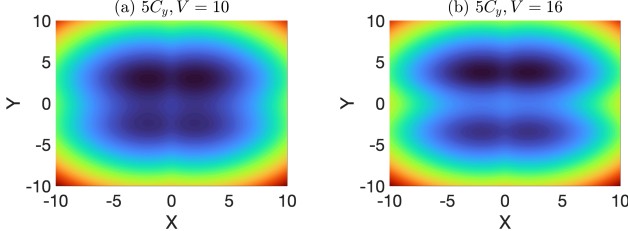

FIG. 8. External potential for tunneling dynamics in the transversely-asymmetric 2D BJJ, described in Eq. 4.4. The transversal asymmetric parameter is $5C_y$ with $C_y = 0.0001$. The barrier heights are (a) $V = 10$ and (b) $V = 16$. The upper part of the space becomes energetically lower with the asymmetric parameter. We show here dimensionless quantities.

$$V'_T(x,y) = \begin{cases} \frac{1}{2}(x+2)^2 + f_T(y), \\ \quad \text{for } x < -\frac{1}{2}, -\infty < y < \infty, \\ \frac{3}{2}(1-x^2) + f_T(y), \\ \quad \text{for } |x| \leq \frac{1}{2}, -\infty < y < \infty, \\ \frac{1}{2}(x-2)^2 + f_T(y), \\ \quad \text{for } x > +\frac{1}{2}, -\infty < y < \infty. \end{cases} \quad (4.4)$$

The asymmetry parameter $C_y$ makes the upper part of the space ($y > 0$) energetically lower, see Fig. 8 for the trapping potential of transversely-asymmetric 2D BJJ.

*a. Many-body survival probability in the transversely-asymmetric BJJ:* We start the investigation of tunneling dynamics in the transversely-asymmetric 2D BJJ using the survival probability in the left side of space, $P(t)$. Fig. 9 presents $P(t)$ for various barrier heights (top to bottom rows, $V = 1, 10, 11, 12, 13,$ and 16) and potential configurations (left to right columns, symmetric and asymmetric with $C_y, 3C_y,$ and $5C_y$). In the many-body dynamics, also shown in the section B, the amplitude of $P(t)$ gradually decays over time, ultimately leading to a density collapse. Here, we illustrate how the rate of density collapse varies with different barrier heights and asymmetries in the potential, revealing faster or slower decay rates depending on the specific conditions.

From Fig. 9, it is evident that the frequency of oscillations of $P(t)$ remains essentially identical for all barrier heights and asymmetries. This invariance arises because both tunable parameters are along the transverse direction and do not alter the Rabi period of tunneling. For $V = 1$, the initial system is fully condensed regardless of the asymmetry introduced in the potential. Consequently, the decay in the amplitude of $P(t)$ exhibits the development of longitudinal fragmentation during the dynamics. The essentially identical rate of decay of $P(t)$ for $V = 1$ across symmetric and transversely-asymmetric BJJs suggests that the rate of density collapse is nearly the same, as the amount of developed longitudinal frag-

mentation is essentially same, unlike observed for the longitudinally-asymmetric BJJs. As the barrier height increases, for example $V = 10$ and 11, and when the system is initially fragmented, the density collapse accelerates with increasing asymmetry in the potential. When the barrier height reaches $V = 12$, the rate of density collapse becomes essentially same to that observed for the fully condensed system at $V = 1$. Further increase of barrier height ($V > 12$), the density collapse of $P(t)$ slows down with increasing asymmetry. Therefore, for a fixed asymmetry in the potential, as the barrier height increases from $V = 10$ to $V = 16$, the rate of density collapse: (i) initially accelerates, (ii) reaches a maximum value, and then (iii) decelerates. Notably, this sequence of processes (i)-(iii) occurs at smaller barrier heights as the asymmetry in the potential increases.

*b. Dynamic occupancy of orbitals and interference of fragmentations in the transversely-asymmetric BJJ:* To understand how asymmetry orthogonal to the junction affects the interference of fragmentations, we present the occupancy of the first excited orbital, $n_2(t)/N$ for selected barrier heights $V = 1, 12, 13,$ and 16 in Fig. 10. The results are shown for asymmetry parameters $C_y$, $3C_y$, and $5C_y$. For comparison, we also include $n_2(t)/N$ for the symmetric potential to highlight the impact of asymmetry. For $V = 1$, the initial system is fully condensed for all asymmetry parameters, as seen in Fig. 7(a). Consequently, during the tunneling dynamics, the first excited orbital retains the shape of the excited $g$-orbital, which remains unchanged throughout the dynamics. Note that here, due to the trap geometry, $y \rightarrow -y$ does not hold. In this case, the occupancy of the excited $g$-orbital monotonously grows, and the asymmetry in the potential does not influence this growth, see Fig. 10(a). When the barrier height is such that the initial state is fragmented, the second orbital is initially the $u$-orbital. In this scenario, the occupancy of the $u$-orbital first decreases, reaches the POM, and then begins to increase. This sequence occurs for all asymmetry parameters. Notably, asymmetry orthogonal to the junction accelerates the system's transition to the POM. This behavior contrasts the results observed for longitudinally-asymmetric BJJs. The transition from a decrease to an increase in the occupancy of the $u$-orbital results from the interference of the longitudinal and transversal fragmentations. Our findings show that the interference of fragmentations strongly depends on the asymmetry orthogonal to the junction. For the symmetric potential, the transition in the occupancy of the $u$-orbital occurs within the barrier height range $V = 7$ to $V = 14$. But for the chosen asymmetry parameters, this range further extends from $V = 7$ to $V = 16$ for the considered interboson interaction. Finally, it is worth noting the behavior of the dynamical occupancy of the ground orbital ($g$-orbital). Our results show that the occupancy of the $g$-orbital gradually decreases over time, irrespective of the asymmetry orthogonal to the junction and the barrier height. However, the rate of this reduction does depend

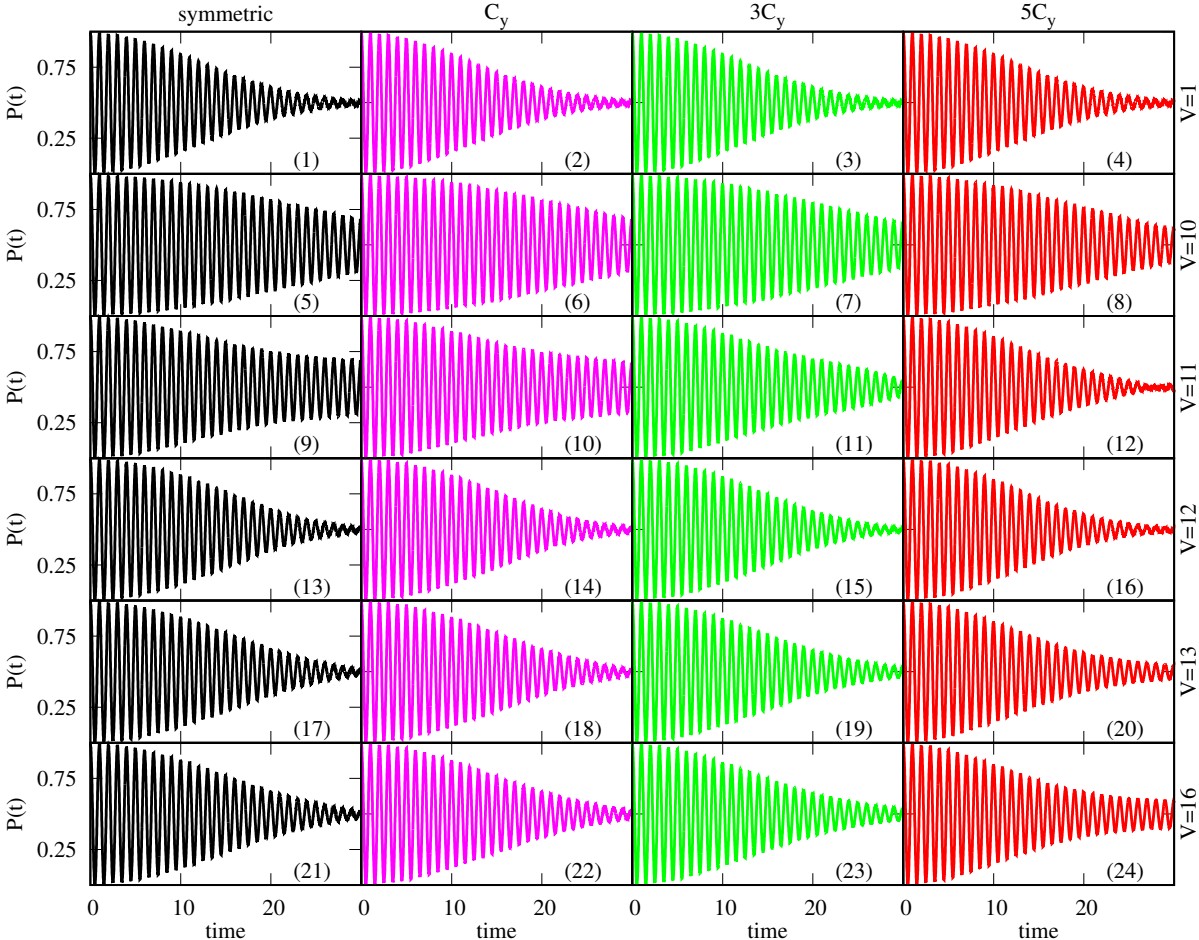

FIG. 9. Dynamics of the many-body survival probability in the left side of space, $P(t)$, for varying barrier heights and asymmetry parameters in the transversely-asymmetric 2D BJJ. From the upper to lower rows present $P(t)$ for growing barrier heights, $V = 1$, 10, 11, 12, 13, and 16, and from the left to right columns show $P(t)$ for symmetric BJJ and asymmetric BJJ with growing asymmetry parameters, $C_y$, $3C_y$, and $5C_y$, respectively, with $C_y = 0.0001$. We show here dimensionless quantities.

on these factors, see [84].

Now we quantify the interference of fragmentations, $\Delta = \dfrac{\text{TF}}{\text{LF}}$, in the tunneling dynamics for the transversely-asymmetric 2D BJJ. The coupling parameter $\Delta$ is plotted as a function of the barrier height $V$ and the asymmetry parameter $C_y$ in Fig. 11 for times $t = 20$ and $t = 30$. $\Delta = 0$ indicates no initial transversal fragmentation in the system. $\Delta \approx 1$ represents maximal coupling, with approximately equal contributions from transversal and longitudinal fragmentations. $\Delta \gg 1$ signifies dominance of transversal fragmentation, where the occupation of the $u$-orbital fails to reach its POM during the tunneling time considered here. Fig. 11 illustrates the temporal evolution of $\Delta$, revealing that the region with $\Delta \approx 1$, corresponding to strong interference, slowly reduces over time. This indicates a dynamical shift in the balance between transversal and longitudinal fragmentations as the system evolves. This behavior is opposite to that observed in longitudinally-asymmetric BJJ.

## V. SUMMARY AND CONCLUSIONS

In conclusion, we have investigated the tunneling dynamics of ground states with varying degrees of fragmentation in two-dimensional asymmetric BJJs, a phenomenon with no mean-field analog. The initial ground state transitions from a condensed state to a two-fold fragmented state depending on the barrier height orthogonal to the tunneling direction. Our findings reveal that asymmetry along the tunneling direction has essentially no impact on the initial transversal fragmentation, whereas asymmetry orthogonal to the tunneling direction reduces the initial transversal fragmentation in the system. As the transversely fragmented ground state begins to tunnel, longitudinal fragmentation emerges dynamically. We explored the interplay between transversal and longitudinal fragmentations for both longitudinally-asymmetric and transversely-asymmetric BJJs. The interplay of fragmentations was analyzed through the collapse of density oscillations in the survival probability,

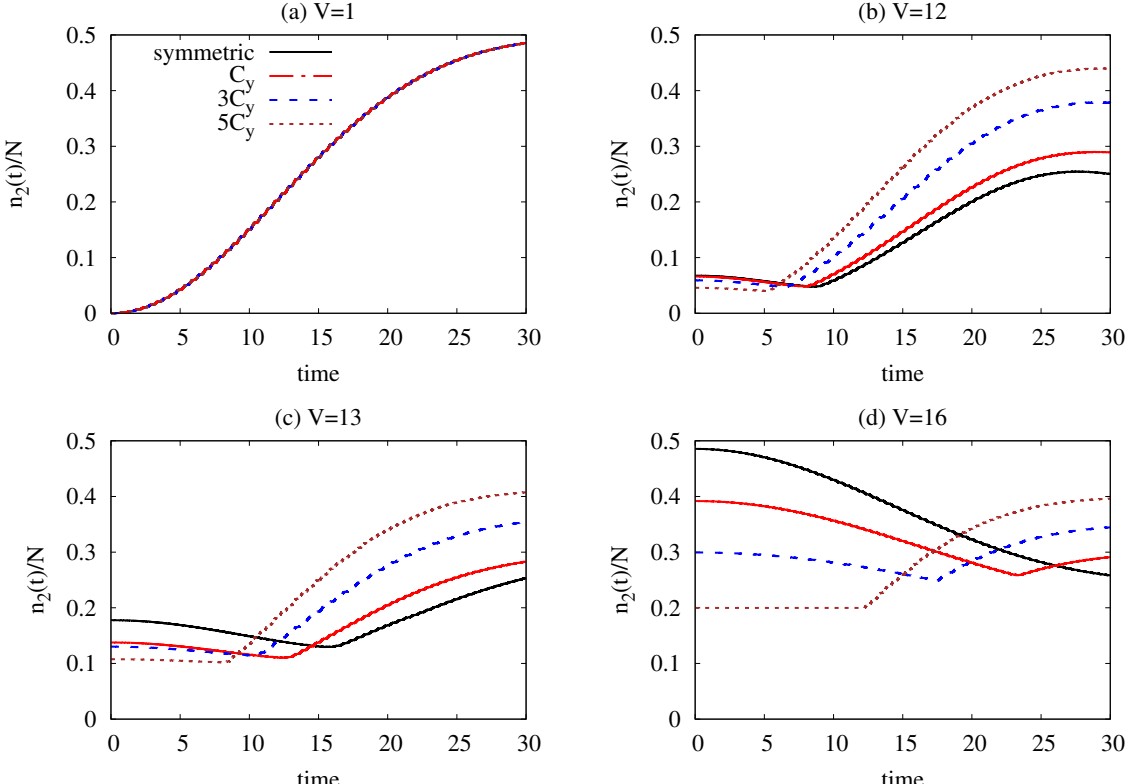

FIG. 10. Time evolution of the occupancy of the first excited orbital, $n_2(t)/N$, for the barrier heights (a) $V = 1$, (b) $V = 12$, (c) $V = 13$, and (d) $V = 16$, in the transversely-asymmetric 2D BJJ. The results are shown for the symmetric BJJ and asymmetric BJJs with selected asymmetric parameters $C_y$, $3C_y$, and $5C_y$ with $C_y = 0.0001$. The monotonously increasing (decreasing) occupancy in the first excited orbital represents strong (weak) longitudinal fragmentation compared to transversal fragmentation see for low barrier height $V = 1$ ( symmetric BJJ with high barrier height $V = 16$). For asymmetric BJJ with high barrier heights, see panels (b), (c), and (d), $n_2(t)/N$ decreases until it reaches a point of minimum. Color codes are explained in panel (a). We show here dimensionless quantities.

the occupation of the first excited orbital, and the normalized uncertainty product.

For longitudinally-asymmetric BJJs, increasing the asymmetry has intriguing impact on the collapse of density oscillations: (i) initially, as asymmetry increases, the collapse slows until the self-trapping condition is reached, (ii) beyond the self-trapping condition, the collapse accelerates until the resonant condition is reached, and (iii) under the resonant condition, the rate of decay of density oscillations increases but remains lower compared to the symmetric BJJ. The collapse rate of the density oscillations is governed by the interference of the transversal and longitudinal fragmentations, quantified by the ratio of reduction in transversal fragmentation and development of longitudinal fragmentation.

Since longitudinal fragmentation evolves over time, the interference of fragmentations is inherently time-dependent. The resonant condition enhances the interference. The self-trapping condition reduces it. The interference of fragmentations also influences the normalized uncertainty product. When the interference is significant, it induces oscillatory behavior in the uncertainty

product. When there is no interference between fragmentations, the uncertainty product remains essentially constant over time.

In the case of transversely-asymmetric 2D BJJs, the behavior of the density collapse exhibits notable dependencies on the barrier height and the degree of asymmetry in the potential. For intermediate barrier height from $V = 8$ to $16$, higher asymmetry in the potential leads to an acceleration of the density collapse. For high barrier height ($V > 12$), increasing the asymmetry results in a deceleration of the density collapse. The observed density collapse is directly linked to the interference of the longitudinal and transversal fragmentations. For symmetric potential, interference of fragmentations is confined to the intermediate barrier height range ($V = 12$ to $14$). Introducing transversal asymmetry in the potential expands the range of barrier heights where the interference of fragmentations occurs.

The results demonstrate that the trap geometry, modulated by tuning the barrier height and asymmetry along and orthogonal to the junction, introduces a wealth of rich dynamical behaviors of fragmented BECs. The ro-

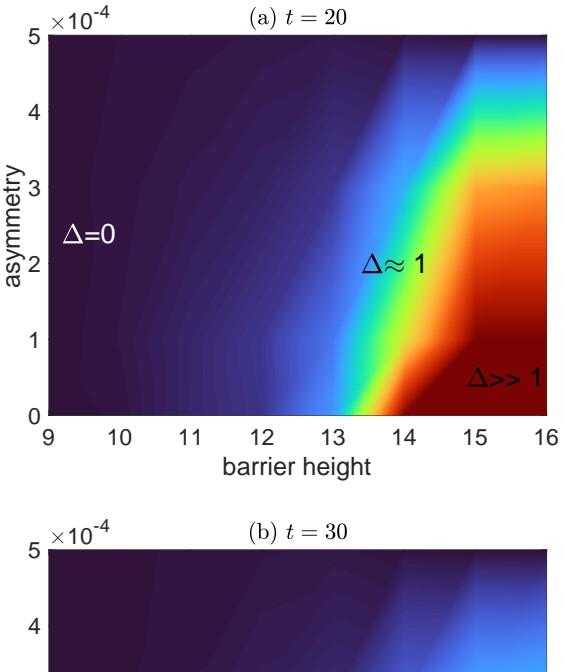

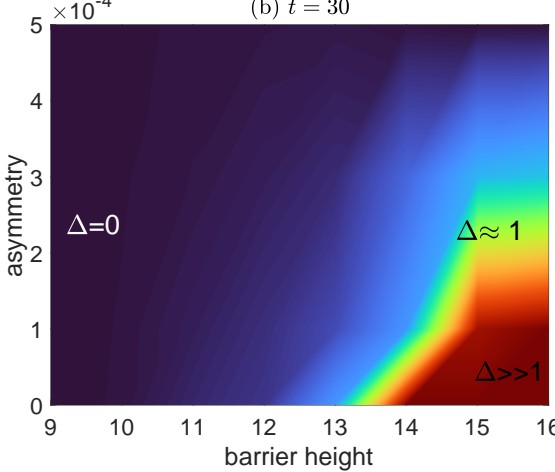

FIG. 11. Interference of transversal and longitudinal fragmentations, $\Delta$, is calculated from the ratio between the reduction of the transversal fragmentation and the development of longitudinal fragmentations $\Delta = \dfrac{\mathrm{TF}}{\mathrm{LF}}$ as a function of barrier height $V$ and asymmetry parameter $C_y$ at time (a) $t = 20$ and (b) $t = 30$. TF and LF are explicitly shown in Fig. 4(c). The results are shown for the transversely-asymmetric 2D BJJ. We show here dimensionless quantities.

**ACKNOWLEDGMENTS**

We acknowledge many insightful discussions with Ofir E Alon. A.B. acknowledges support from the U.S. NSF through Grant Number PHY-2409311. SKH acknowledges the support from the Department of Science and Technology (DST), India, through TARE Grant No.: TAR/2021/000136. Computation time on the High-Performance Computing system Hive of the Faculty of Natural Sciences at the University of Haifa, and the Hawk at the High Performance Computing Center Stuttgart (HLRS) are gratefully acknowledged.

bustness of the interference of fragmentations with respect to asymmetry in the 2D BJJ is observed. The inclusion of asymmetry and barrier height in the two-dimensional BJJ framework, as presented here, can be extended to explore the dynamics of various exotic quantum phases, such as supersolids [15, 88, 89], superfluid and Mott insulator [90], and bosonic and fermionic Tonks-Girardeau gases [91, 92]. Furthermore, the many-body physics discussed in this work is expected to have implications for the broader scientific community, particularly in fields such as atomtronics [65] and precision metrology [93].

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

## SUPPLEMENTAL MATERIAL

In this supplemental material, we provide additional details to complement the main text. Specifically, we present the dynamical occupation of the ground orbital for both longitudinally-asymmetric and transversely-asymmetric two-dimensional (2D) bosonic-Josephson junctions (BJJs). The time evolution of the many-body uncertainty product for the transversely-asymmetric 2D BJJ is also discussed. Furthermore, we demonstrate the convergence of our results with respect to time-adaptive orbitals and the number of grid points used in the computations.

### A. Dynamical occupation of the ground orbital for the longitudinally-asymmetric and transversely-asymmetric 2D BJJs

The main text thoroughly discusses the occupation of the $u$-orbital and briefly touches on the $g$-orbital. Here, we focus on the dynamical occupation of the $g$-orbital, $n_1(t)/N$, for barrier heights $V = 1$, 12, and 16. For each barrier height, we present results for both symmetric 2D BJJs and longitudinally-asymmetric 2D BJJs with selected asymmetry parameters $C_x = 0.01$ and $25C_x$ (see Fig. S.1).

Since the chosen longitudinal asymmetry does not affect the initial conditions in this study, we observe that, for a fully condensed system with $V = 1$, $n_1(t)/N$ begins

at a value of one for all asymmetry parameters. Similarly, for a partially fragmented system with an intermediate barrier height of $V = 12$, the initial $g$-orbital occupation is approximately 0.94, while for a fully fragmented system, it starts around 0.5.

In the fully condensed system, the $g$-orbital occupancy decreases monotonically for all asymmetry parameters, with the maximum rate of decay observed for the symmetric 2D BJJ. A similar trend is seen for partially fragmented and fully fragmented initial systems, where the decay rate of $n_1(t)/N$ is again maximal for symmetric potentials. Interestingly, for $V = 12$ with asymmetry $C_x$, the $g$-orbital occupation initially increases before decreasing. This behavior is characteristic of intermediate barrier heights and occurs across all asymmetry potentials, except under resonant conditions.

Similarly, we analyze the occupancy of the $g$-orbital for the transversely asymmetric 2D BJJs at barrier heights $V = 1$, 12, and 16. For each barrier height, we select asymmetry parameters $C_y = 0.0001$, $3C_y$, and $5C_y$.

For the condensed initial system at $V = 1$, $n_1(t)/N$ starts at a value of one for all asymmetry parameters. In contrast, for the larger barrier heights $V = 12$ and $V = 16$, the initial system is fragmented to varying degrees, with the fragmentation level depending on the asymmetry parameter. In general, we observe that $n_1(t)/N$ decreases monotonically over time for all barrier heights and asymmetry parameters. For the condensed system, the dynamical occupation of the $g$-orbital is independent of asymmetry along the $y$-direction. Interestingly, it is found that transverse asymmetry, orthogonal to the tunneling direction, enhances the rate of decay of the $g$-orbital occupation. This behavior contrasts the results observed for longitudinally-asymmetric potentials.

## B. Time evolution of the many-body uncertainty product for the transversely-asymmetric 2D BJJ

In this section, we examine the effect of the coupling between longitudinal and transverse fragmentations on the many-body normalized uncertainty product along the $y$-direction, $U(t)/U(0)$ where $U(t) = \frac{1}{N^2}\Delta_{\hat{Y}}^2(t)\Delta_{\hat{P}_Y}^2(t)$. Figure S.3 illustrates $U(t)/U(0)$ for selected barrier heights $V = 1$, 9, 10, 12, 13, and 16. For each barrier height, we present results for both symmetric 2D BJJs and transversely-asymmetric 2D BJJs with asymmetry parameters $C_y = 0.0001$, $3C_y$, and $5C_y$.

For a fully condensed system ($V = 1$), there is no initial transverse fragmentation, and only longitudinal fragmentation develops during the tunneling process for both symmetric and asymmetric 2D BJJs. Consequently, $U(t)/U(0)$ remains essentially constant throughout the dynamics. As the barrier height increases and the initial system becomes transversely fragmented, the competition between longitudinal and transversal fragmentation introduces oscillatory behavior in $U(t)/U(0)$. This is ev-

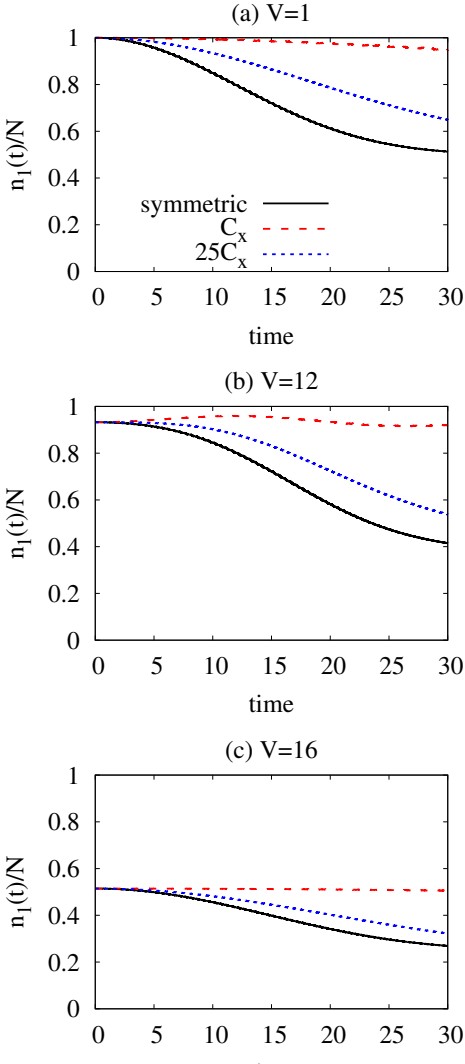

FIG. S.1. Time evolution of the occupancy of the ground orbital ($g$-orbital) for the barrier heights (a) $V = 1$, (b) $V = 12$, and (d) $V = 16$. In each panel, results are shown for symmetric 2D BJJ and the longitudinally-asymmetry 2D BJJ with asymmetry parameters are, $C_x$, and $25C_x$ with $C_x = 0.01$. Color codes are explained in panel (a). We show here dimensionless quantities.

ident in Figure S.3(b) for $V = 9$ and Figure S.3(c) for $V = 10$. The amplitude of these oscillations is maximal for the symmetric potential and minimal for the highest asymmetry parameter ($5C_y$), as clearly shown in Figure S.3(c). With further increases in the barrier height, as seen in Figure S.3(d) for $V = 12$ and Figure S.3(e) for $V = 13$, the coupling between longitudinal and transverse fragmentations becomes strong, leading to an increase in magnitude of $U(t)/U(0)$ over time. Finally, for $V = 16$ (Figure S.3(f)), the initial transverse fragmentation dominates over the longitudinal fragmentation developed during the tunneling process. This weakens the coupling between the two types of fragmentations, result-

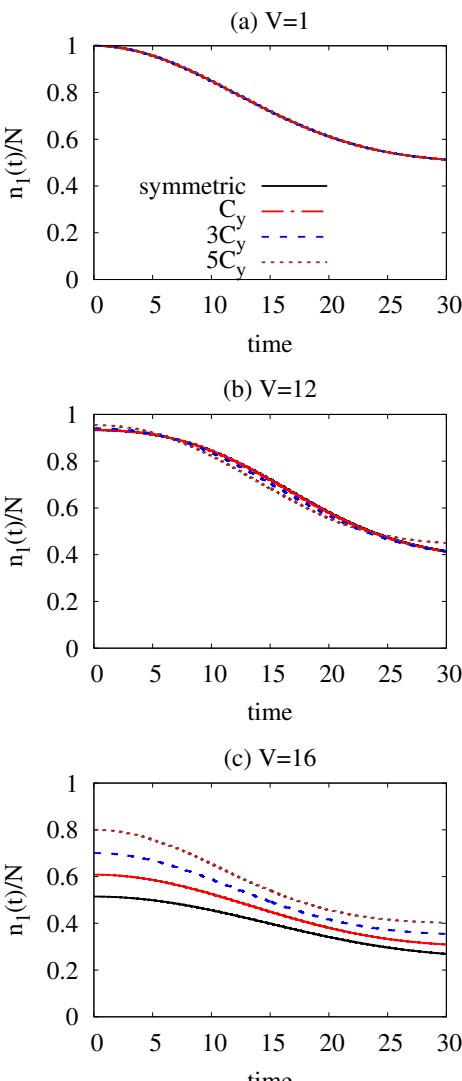

FIG. S.2. Time evolution of the occupancy of the ground orbital ($g$-orbital) for the barrier heights (a) $V = 1$, (b) $V = 12$, and (d) $V = 16$. In each panel, results are shown for symmetric 2D BJJ and the transversely-asymmetry 2D BJJ with asymmetry parameters are, $C_y$, $3C_y$, and $5C_y$ with $C_y = 0.0001$. Color codes are explained in panel (a). We show here dimensionless quantities.

ing in a nearly frozen dynamical behavior.

## C. Convergence of results for the longitudinally-asymmetric and transversely-asymmetric 2D BJJs

The multiconfigurational time-dependent Hartree for bosons (MCTDHB) method is employed in this work to compute the ground (initial) state, which becomes fragmented depending on the barrier height $V$. The many-body Hamiltonian is represented using $128^2$ exponential discrete-variable-representation (DVR) grid points within a box of size $[-10 \times 10) \times [-10 \times 10)$ to compute the ground state and its subsequent real-time propagation. In the main text, all many-body quantities were calculated using $M = 8$ time-adaptive orbitals for both longitudinally-asymmetric and transversely-asymmetric 2D BJJs. Here, we demonstrate the convergence of the quantities discussed, showing that the time-dependent many-boson wavefunction built with $M = 8$ time-adaptive orbitals yields numerically converged results. Specifically, we verified convergence by comparing results with $M = 10$ time-adaptive orbitals on a $128^2$ DVR grid and with $M = 8$ time-adaptive orbitals on an increased grid density of $256^2$ DVR points for all barrier heights and asymmetry parameters. We show convergence for the most sensitive quantities discussed in the main text, the occupation numbers and the uncertainty product.

To illustrate convergence, we focus on a barrier height of $V = 12$ and the resonant condition $(25C_x)$ for the longitudinally-asymmetric 2D BJJ. For the transversely-asymmetric 2D BJJ, we consider the maximal asymmetry $(5C_y)$. We present the occupation of the $u$-orbital, $n_2(t)/N$, and the normalized uncertainty product along the $y$-direction, $U(t)/U(0)$, see Fig. S.4 for longitudinally-asymmetric 2D BJJ and Fig. S.5 for transversely-asymmetric 2D BJJ. The overlapping curves for all quantities, with increased grid density and a higher number of orbitals, confirm that the results presented in this work are converged for $M = 8$ time-adaptive orbitals with $128^2$ DVR grid points in a box of size $[-10 \times 10) \times [-10 \times 10)$.

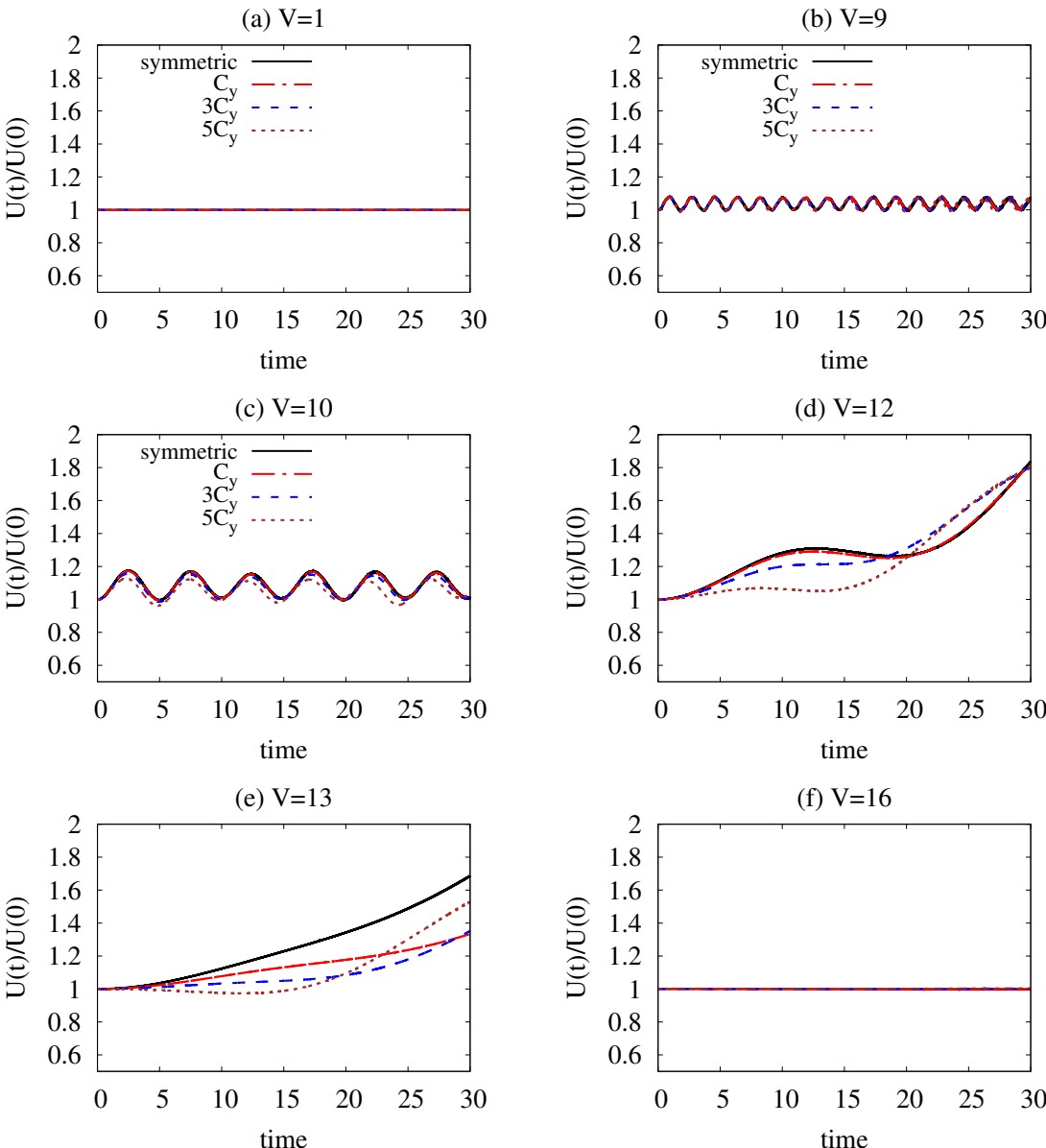

FIG. S.3. Time evolution of the many-body normalized uncertainty product along the transverse direction $U(t)/U(0)$ in the symetric and transversely-asymmetric 2D BJJ. The barrier heights are (a) $V = 1$, (b) $V = 9$, (c) $V = 10$, (d) $V = 12$, (e) $V = 13$, and (f) $V = 16$. In each panel, $U(t)/U(0)$ is shown for the asymmetry parameters, $C_y$, $3C_y$, and $5C_y$ with $C_y = 0.0001$. Color codes are explained in the top row. We show here dimensionless quantities.

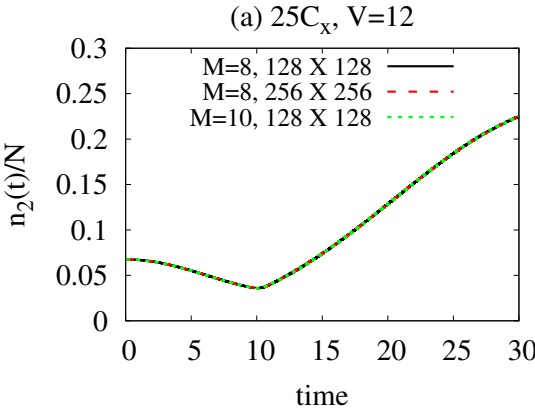
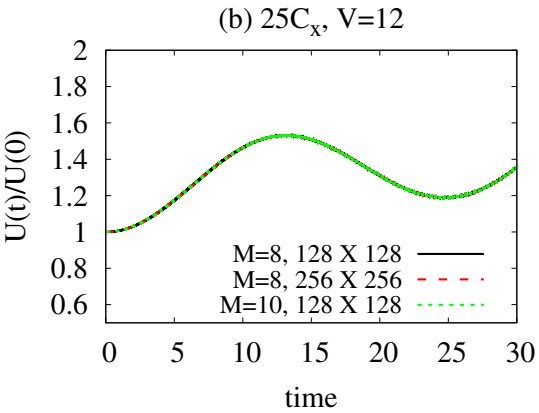

FIG. S.4.    Convergence of the quantities, (a) the dynamical occupation of the $u$-orbital, $n_2(t)/N$, and (b) the normalized uncertainty product along the transverse direction, $U(t)/U(0)$ for the longitudinally-asymmetric BJJ. The barrier height is $V = 12$ and the asymmetry parameter is $25C_x$ with $C_x = 0.01$. Convergences are verified using $M = 10$ time-adaptive orbitals with $128^2$ grid points and $M = 8$ time-adaptive orbitals with $256^2$ grid points. We show here dimensionless quantities Color codes are explained in each panel. We show here dimensionless quantities.

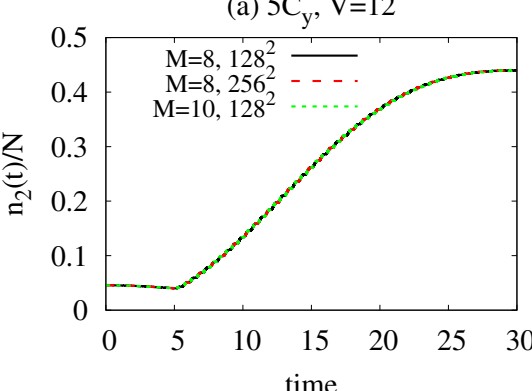
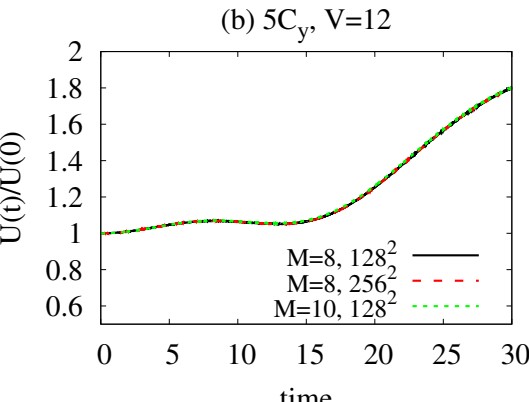

FIG. S.5.    Convergence of the quantities, (a) the dynamical occupation of the $u$-orbital, $n_2(t)/N$, and (b) the normalized uncertainty product along the transverse direction, $U(t)/U(0)$ for the transversely-asymmetric BJJ. The barrier height is $V = 12$ and the asymmetry parameter is $5C_y$ with $C_y = 0.0001$. Convergences are verified using $M = 10$ time-adaptive orbitals with $128^2$ grid points and $M = 8$ time-adaptive orbitals with $256^2$ grid points. We show here dimensionless quantities Color codes are explained in each panel. We show here dimensionless quantities.