# Peer review of "Interplay of asymmetry and fragmentation in the many-body tunneling dynamics of two-dimensional bosonic Josephson junctions"

_SciPost Physics_

## Round 1 · Referee Report · Romain Dubessy (Referee 1) · 2025-7-17

Strengths

1- relies on a well established and robust numerical method 2- figures are clear 3- complete bibliography

Weaknesses

1- the significance of some results is unclear 2- the analysis is mostly qualitative 3- the work may appear as incremental

Report

The manuscript entitled "Interplay of asymmetry and fragmentation in the many-body tunneling dynamics of two-dimensional bosonic Josephson junctions" by Anal Bhowmik and co-authors reports on a numerical study of the dynamics in a asymmetric two-dimensional bosonic Josephson junction, using a many-body method (so called MCTDHB). This work follows a series of works by some of the authors, studying roughly the same geometry, with the same methods and using the same analysis. The main novelty here is the addition of a tilting potential either in the longitudinal or transverse direction (with respect to the junction axis). As a consequence part of the results presented for reference are already present in previous publications. For example, the "symmetric" case is already fully described in Ref. 59. Similarly, the asymmetric one dimensional Josephson junction (corresponding to low V values in the present work) was studied in Ref. 33.

To my opinion, even if the present work is different from previously published result because there are new parameters, it is difficult to evaluate the novelty and importance of the results, for the following reasons:

1) The results are not compared with existing literature or simple analytical estimates. For example I find it not surprising at all that the initial fragmentation does not depend on the longitudinal asymmetry ($C_x$), as it results only in a translation of the initial many body wavefunction (this "result" is highlighted in the conclusion). Similarly I find the choice of parameters is poorly commented (except for the self trapping condition $C_x=0.1$).

2) The part "III Quantities of interest" does not introduce new concepts and should be placed as an annex. It could be replaced by a short discussion of what is already known for $P(t)$ and $U(t)$ based on previous works, and a justification of why those quantities are interesting in this context.

3) The discussion of the different trap potentials is difficult to follow. Instead of equations 4.1, 4.2, 4.3 and 4.4, the authors could define the initial trap $V(x,y)=\frac{(x+2)^2}{2}+\frac{y^2}{2}+V e^{-y^2/8}$ and the junction trap $V_J(x,y)$ (with the polynomial interpolation) and specify that in section IV.A they add an extra $-C_x x$ "tilting" potential ($-C_y y$ in section IV.B).

4) The discussions are mostly qualitative: for example the authors discuss the decay rate of $P(t)$ as a function of the amplitude of $V$, but they do not give a definition of this rate. As the variations of this rate seem rather small (see Fig. 3) it seems important to define it properly.

5) I am not convinced by the figure of merit $\Delta$ introduced for the first time in this work. In particular I have a problem of methodology: how it is defined when the minimum of the oscillation is not visible in the curve (Fig 4.d for example) ? Moreover I failed to understand the added value of this parameter with respect to $P(t)$, $U(t)$ or $n_2(t)/N$... This should be discussed and emphasized.

6) Finally the authors added some control parameters to a many-body Hamiltonian and find that the dynamics depend on those parameters. This I could have guessed beforehand. So what do we learn about the initial model ? Does it means that in an eventual experiment we need to control very well the tilt of the traps ? Does it prevents to use the setup as a Bosonic Josephson junction ? Does it opens new perspectives ? I feel that these points should be addressed in the discussion and in the conclusion.

For these reasons I do not think that the manuscript in the present form meets the acceptance criteria of SciPost Physics. I have also several minor remarks that I detail in the requested changes.

Requested changes

page 2: "By systematically varying the asymmetry..." Why the authors have not considered negative asymmetries ? In Ref. 33 $C_x$ reaches larger values, why set the limit to $C_x=0.25$ here ?

page 2: What is the role of the parameter $\Lambda$ ? It does not appear elsewhere in the manuscript... The value of $\lambda_0$ is not given so the results cannot be reproduced. Please clarify.

page 3: Move the technical parts of section III in an appendix and discuss the role of each computed quantity. Why do the authors restrict the study to the population of the second orbital (Ref 4. suggests that other orbitals may also exhibit interesting dynamics) ?

page 4: I do not find Figure 2 particularly useful, it is worth trying a different coloring (same remark for Figure 8).

page 4: "asymmetry parameters $C_x$, 10$C_x$, ...". I have a problem with the way the authors report the values of $C_x$. $C_x$ is a parameter of the model so it should be reported as $C_x=0.1$, $C_x=0.2$, and so on (as they did for the parameter $V$ for example). Please correct, the multiple occurrences throughout the paper. Same remark for $C_y$.

page 4: "The geometry of the trap dictates... along the $y$-direction." I wonder if this explanation still holds for large barrier ($V=16$), because later the authors mention that for large barriers the system is equivalent to two independent condensates, each localized in a well. In that case I expect two degenerate orbitals, each one localized in one well: then the notion of parity is not appropriate anymore...

page 5: In the time dependent Schrodinger equation I believe the dot between $r_1$ and $r_2$ should be a comma ?

page 6: Concerning figure 3 and surrounding text. The vertical axis ticks are not consistent across the panels, please check. The period of oscillation seems to depend strongly on $C_x$, can it be explained ? Please define properly the decay rate. It is not clear from Fig.3 and Fig. 4 how the authors conclude that "Moreover it is found that for a fixed barrier height, the density collapse is slower due to the slower development of longitudinal fragmentation". How the rate of growth of longitudinal fragmentation is defined ?

Page 7-8: see my general remarks about the role of $\Delta$.

page 12: Again the decay rate of $P(t)$ should be defined properly. As all the curves display qualitatively the same behaviour for this section it should be easier to do. A figure showing how the decay rate changes with $V$ or $C_y$ would be much more informative !

Recommendation

Ask for major revision

---

## Editorial Decision

awaiting_resubmission